# Mobile Internet Technology Adoption for Sustainable Agriculture: Evidence from Wheat Farmers

Nawab Khan [1], Ram L. Ray [2], Hazem S. Kassem [3] and Shemei Zhang [1],*

1 College of Management, Sichuan Agricultural University, Chengdu 611100, China; nawabkhan@stu.sicau.edu.cn
2 College of Agriculture and Human Sciences, Prairie View A&M University, Prairie View, TX 77446, USA; raray@pvamu.edu
3 Department of Agricultural Extension and Rural Society, King Saud University, Riyadh 11451, Saudi Arabia; hskassem@ksu.edu.sa
* Correspondence: 14036@sicau.edu.cn

**Abstract:** Mobile internet technology (MIT) is considered a significant advancement in information and communication technology (ICT), due to its crucial impact on the financial system and social life. In addition, it is an essential technology to overcome the digital divide between urban and rural areas. In terms of agricultural advancement, MIT can play a key role in data collection and the implementation of smart agricultural technologies. The main objectives of this study were to (i) investigate MIT adoption and use in sustainable agriculture development among selected wheat farmers of Pakistan and (ii) examine the crucial factors influencing MIT adoption. This study selected 628 wheat farmers from four districts of Khyber Pakhtunkhwa Province (KPK), Pakistan, for sampling. This study used a bivariate probit method for sampling wheat farmers. The analysis of wheat farmer's data showed farmer's age, farm size, farm location, and knowledge about Internet technology (IT) are strongly correlated with MIT adoption in sustainable agriculture development. Results showed on average, 65% of wheat farmers have mobile devices supporting these Internet technologies, and 55% use MIT in agricultural environments. Since the extant research on MIT adoption for agriculture production in Pakistan is sparse, this study helps advance MIT adoption-based studies. These outcomes may draw the attention of decision-makers dealing with IT infrastructure and agricultural equipment who can support farmers adopting MIT.

**Keywords:** sustainable; agriculture; rural; mobile internet technology; bivariate probit model; agricultural modernization; Pakistan; wheat; smart

## 1. Introduction

Agriculture is not only a source of food but also a source of employment and opportunities, especially in rural areas. Therefore, the development and growth of agriculture, significantly advancing technology, must be highly prioritized, especially in developing countries [1,2]. This is because food security is one of the main goals of the governments of these developing countries. It is worth noting that the United Nations defined food security as all people, at all times, having physical, social, and economic access to sufficient, safe, and nutritious food, meeting their food preferences and dietary needs for an active and healthy life [3]. Agriculture plays a pivotal role in poverty alleviation and economic development. The link between agricultural productivity growth and poverty reduction is well documented by Raza et al. [4]. Advancing the agricultural system is critical to ensure food security and reduce poverty. However, in developing countries, small farmers frequently encounter several challenges that prevent them from advancing the agricultural system. These barriers include, for instance, asymmetric information regarding importing traders and exporting consumers, high transaction expenses, insufficient agricultural services, and insufficient access to credit resources [5,6]. Especially due to information

asymmetry (when one party or group, or community has more information than others), small farmers, particularly rural farmers, are not equally informed and equipped with advanced technologies which help advance their skills and knowledge of farming [7]. Due to a lack of appropriate training and skills, they may not be able to adopt/use the latest equipment such as digital technologies (e.g., mobile phone, internet, and computer) and inputs (e.g., seeds, fertilizers, fungicides, and pesticides or effective use of existing agricultural inputs) to increase agricultural productivity [8]. Therefore, the crop yields and incomes of these farmers are low, which is not conducive to their livelihoods and rural development [9–12]. Hence, advanced methods used to reduce information asymmetry are worthwhile, especially when using information and communication technology (ICT) to improve farm performance and overall agricultural productivity.

Technological innovations are becoming increasingly crucial in agricultural development and productivity. The use of ICTs in agriculture provides a more efficient and cost-effective way of sharing and exchanging knowledge more widely. Farmers benefit from access to key information such as pest reports, weather conditions, and market prices [13]. The use of Internet technology (IT) can help reduce information asymmetry as it disseminates information fast and at a low cost. Mobile phones have a long history of only being used for voice communication and text message. Recent advancements in mobile phones, including smartphones, revolutionized using the internet on mobile phones. The essential primary features of mobile phones allow users to access the internet without using a computer. Internet access through a mobile phone influenced us, including farmers [14]. Previous evidence has proven that the adoption/use of mobile Internet technology (MIT) can improve the accessibility of financial and agricultural services for smallholder farmers [8,15], the availability of input and output markets [16], and the promotion of income activities (such as non-agricultural commodities). Due to the significant advantages of MIT, some developing countries have adopted many Internet-based agendas to implement farms better for rural development [17–19]. For example, the internet + agriculture + finance model, rural e-commerce, and farmer's field school are Internet-based agendas used in China [20].

The ICT has drastically modified communication, sales and information methods [21]. Several studies have investigated the effects of Internet use by ICTs, such as mobile phones and computers on-farm performance and rural household's productivity income [22–24]. These studies focused on the selection bias of IT used through the application of various methods such as instrumental variables (IV), endogenous therapy regression (ETR), and propensity score matching (PSM) models [20]. By estimating the PSM model, Issahaku et al. [25] found that mobile phone use significantly improves agricultural productivity in Ghana. Ma et al. [26] estimated an ETR model and showed that Internet use significantly increases household income and expenditure in rural China. Kelemu [27] found a significant impact of mobile phones and mobile communications on improving wheat productivity and efficiency in Ethiopia. In addition, Quandt et al. [28] examined perceptions about the effects of mobile phone use on agricultural productivity in rural Tanzania, East Africa. They reported that about 47% of respondents stated that mobile phone usage had reduced the amount of time they spent buying inputs or selling crops, and 50% reported that mobile phone usage had reduced the amount of money they spent on-farm activities. Further, 64% reported that the mobile phone usage has increased profits from farming compared to when the respondent did not have a mobile phone.

The adoption/use of MIT may affect the production of crops because it may influence farmer's production behaviors in combining and using different inputs (e.g., labor, capital assets, fertilizers, and pesticides) [29]. Technical efficiency refers to the ratio of farmer's observed output to the maximum realizable output given the existing inputs [30], indicating the use efficiency of different agricultural inputs. Several studies show that the adoption/use of MIT will significantly affect the behavior of farmers to use seeds and fertilizers [18] and land expansion [31]. To the best of our knowledge, apart from work completed for Zambia by Mwalupaso et al. [32], sparse findings have investigated

the influence of MIT use on crop production. Mwalupaso et al. [32] studied the impact of mobile phone access to IT on corn production in Zambia. They observed that mobile phones have greatly improved the technical efficiency of farmers.

This article aims to examine MIT adoption by wheat farmers in Pakistan. A bivariate probit model is employed using a representative data set of 628 wheat farmers from four districts of Khyber Pakhtunkhwa (KPK) province in Pakistan for sample selection. We are concerned about wheat farmers for two purposes. First, Pakistan is the eighth leading producer of global wheat production, after China, India, Russia, the United States, Canada, Australia, and Ukraine (Figure 1). Despite the high total yield, Pakistan's wheat yield is only three tons per hectare, ranking 15th globally [33]. Since lower wheat yield is reducing the farmer's income and weakening the effectiveness of the Pakistani wheat industry in the global market, the development of wheat production must be promoted. Secondly, Pakistan's broadband Internet has developed rapidly in recent years, especially in remote areas. Currently, Internet users using any device have reached 76.38 million. This means that 35% of the population is using IT. The use of the Internet plays an important role in farmer's lives and agricultural production. In addition, technological innovation has greatly shaped agriculture. Humans have developed new ways to make farming more efficient and grow more food [29]. As mentioned above, due to the urgent need to increase wheat productivity, this research can reveal whether the adoption/use of the MIT can affect farmer's decision to utilize inputs wisely to increase wheat yield in Pakistan.

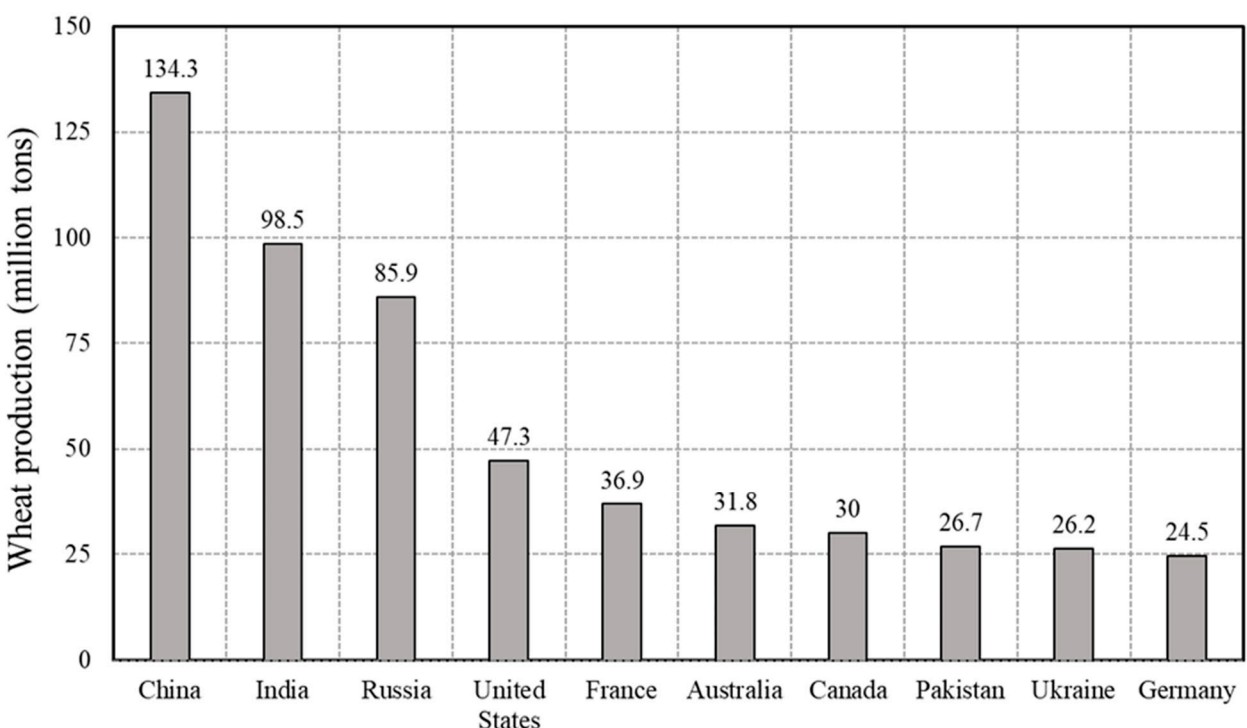

**Figure 1.** Total wheat production in the top ten countries in 2021. (Data source: FAOSTAT).

In order to determine the important factors that influence the MIT adoption in agricultural advancement, it is assumed that this set of factors can affect the mobile devices (MDs) adoption, including farmer's characteristics, MIT, and farm characteristics. Moreover, factors that influence the knowledge provided by MIT, which may be useful to decision makers and companies, are also identified. This information can be implemented in the marketing policies of agricultural systems that depend on MIT and equipment such as precision and intelligence. The main objectives of this study were to (i) investigate MIT adoption and use in sustainable agriculture development among selected wheat farmers of Pakistan and (ii) examine the crucial factors influencing MIT adoption.

The remainder of the article is organized as follows. Section 2 provides various study hypotheses based on a review of extant literature. Section 3 introduces the model specification, study area, sampling strategy, and data collection. Section 4 describes the results and discussion of the study hypothesis. Section 5 presents the research conclusions.

## 2. Literature Review

A growing number of researches focus on the impact of ICTs on farmers. The dissemination and adoption of MIT in developing countries have attracted the attention of economists. The broad growth in MIT advancements over the past decade offers innovative prospects to overcome these search and deal prices and the potential to enhance agricultural productivities and incomes [34,35]. In particular, the theory proposes that MIT affects many of the main channels through which smallholder's incomes are affected. First, MIT may reduce the search cost for farmers, obtaining price information in more markets and selling in the market with the highest price after deducting transportation costs [36,37]. Second, improved access to information may enhance the bargaining position of farmers with traders in the absence of sales in different markets [38]. Third, MIT may allow farmers to use mobile phones to complete sales, thereby reducing the uncertainty associated with sales in remote markets [39]. Fourth, if information technology increases farmer's commodity prices, and since agriculture is price elastic, this will increase the future production of such commodities [40,41].

From an arbitrage perspective, past studies have shown that if farmers could obtain selling price information in other markets through ICT, and the transportation cost is lower than the price difference between these two markets, farmers would go to other markets to sell their products for better profits. As a result, arbitrage that occurs between markets will decrease, and Pareto efficiency will be achieved [34,42]. Along this line, Jensen [43] observed the mobile phone's role in market arbitrage in local agricultural markets in Indian states. The key outcomes suggest that the introduction of mobile phones stimulated cross-market arbitrage, reduced price differences across markets, and eliminated the oversupply of sardines in individual local markets. As a result, both the producer's profit and the consumer's surplus increase. Meanwhile, Aker [39] examined mobile phones' impact on the grain market in Niger and initiated that price differences between markets have decreased. A significant additional study looked at the ICT impact on agricultural development and market choices, focusing on mobile internet and radio technology [34].

Especially in developing nations, farmers often can sell their products to merchants who shuttle between villages and markets or transport produce to the nearest market themselves. Uncertainty about market prices is usually high for farmers due to remote communities and poor market communication. Courtois and Subervie [44] demonstrated the conditions under which market information services are beneficial to farmers and examined efficiency issues related to information asymmetry. The causal impact of the mobile phone market information services program on farmer's marketing performance in Ghana showed that farmers who benefited from a market information service program received significantly higher prices for peanut and maize: about 7% higher for peanut and 10% higher for maize than what they could have received had they not participated in the market information services program. Additionally, Zanello [45] used a new dataset of 393 households in northern Ghana with comprehensive information on market transactions, agricultural development, and ICT use through a transaction cost context. The results showed that obtaining market info through mobile technology had a positive and substantial influence on market participation, with a better effect on households with surplus food crops and agricultural progress.

In China, the advancement of modern technology increased China's rural non-farm employment by 10–20% and ultimately upsurged the income of rural people [34]. In conclusion, the expansion of ICT in developing countries provides new technologies and opportunities for farmers to access information. Remarkably, the rapid development of computers and MIT have introduced a new search technology that offers numerous

advantages over other alternatives in terms of cost, geographic coverage, etc. Compared to broadcasts and newspapers with one-way communication systems, information can be obtained more efficiently using modern information systems. Especially in developing countries, the rapid growth of MIT access and computers have ushered in an era in which everything is changed by information technology, which has an equal effect on rural farmers who have traditionally been far away from the development of information technology [46].

Specifically, timely access to market information through communication networks can help farmers decide what crops to grow, where to sell their products, and how to improve input efficiency. ICTs can also provide unprecedented access to rural finance, while financial and information service networks can provide microfinance opportunities for locals and small businesses [34,46–48]. As mentioned earlier, statistics and some empirical studies can provide general patterns for mobile telegraphy. However, research on mobile Internet technology development patterns and their consequences based on raw data is particularly significant.

## 3. Study Hypothesis for the MIT Adoption

According to Savari and Gharechaee [49], the widespread dissemination of innovation theory has been used in some scientific disciplines to describe advanced technology adoption through individuals, society, or organizations. This hypothesis reflects many variables expected to affect the adoption of modern technologies. The variable set contains adapters and innovations as well as firm characteristics. To the best of our knowledge, there is a lack of understanding of MIT used in agricultural development among small farmers in developing countries, including Pakistan. Therefore, these assumptions are also generally considered from the literature on adopting information technology and MIT. The study also reviewed the literature on farmers' adoption/use of computers and the Internet.

### 3.1. Farmer's Characteristics

Savari and Gharechaee [49] analyzed the MIT by American farms and determined that the adoption rate of the Internet and computers is declining with age. Indeed, as the descriptive statistics show, young people have a higher Internet access rate. Bort-Roig et al. [50] found that the use of MIT is higher among young people. Therefore, in terms of the adoption and usage of MIT, it is not surprising that some studies have shown that the adoption rate decreases with age [51]. Therefore, the following conditions are assumed:

**Hypothesis 1a.** *(Age): Older farmers are less likely to adopt MIT.*

Gender plays a crucial role in information technology adoption decisions [49,52]. Compared with male, female farmers are less likely to adopt modern technologies [49,53]. However, when it comes to adopting the internet for agriculture, the results are mixed. For example, Mendes et al. [54] found no association between gender and Internet adoption, while Khan et al. [29] found that gender was correlated with internet adoption. Specifically, male farmers are more likely to become MIT adopters. Haq et al. [55] pointed out that more and more men use MDs to access the internet. Regarding the adoption of MIT, Mayzelle et al. [56] found that the adopter is more likely to be male. Hence, the following assumption was made:

**Hypothesis 1b.** *(Gender): Male farmers are more likely to adopt MIT.*

Education develops the ability of individuals to understand, learn and adopt new technologies, such as the internet [29], and therefore is a significant socio-economic factor for information technology adoption [57–59]. Based on this insight, Fabregas et al. [60] pointed out that more formal schooling is optimistically associated with the use of IT in agricultural development. Generally speaking, well-educated adults are more likely to access the internet [20]. In addition, MIT adoption and education are also positively

correlated [23]. Additionally, Nie et al. [61] believed that education is an important factor influencing IT adoption. Therefore, the following assumption was made:

**Hypothesis 1c.** *(Education): farmers with higher education are more likely to adopt MIT.*

Innovation is manifested in the willingness to check the latest technology [62] and is an important factor affecting the latest technology adoption. Haile et al. [53] revealed that innovation is positively associated with precision agriculture adoption in the agriculture field. Regarding MIT, Singh et al. [63] pointed out evidence that innovation is associated with MIT adoption. Therefore, the following assumption was made:

**Hypothesis 1d.** *(Innovativeness): More advanced farmers are more likely to adopt MIT.*

*3.2. Mobile Internet Technology Characteristics*

Sekabira and Qaim [64] indicated that some farmers did not utilize IT for safety reasons. In addition, Rehman et al. [65] pointed out that overall, security issues are the main obstacle to the adoption of IT. O'Leary et al. [66] found that security and privacy risks are the main issues when using mobile information services. For example, IT can spread diseases, where hackers can intercept signals, thereby imperiling the safety of individuals' data transmitted. Consequently, the following assumption was made:

**Hypothesis 2.** *(Awareness of IT risks): farmers aware of IT risks are unlikely to adopt MIT.*

*3.3. Farm Characteristics*

Because of several factors, such as the high demand for information on the farm and the complexity of the organization, large farm size is positively associated with MIT adoption [67]. Generally, the size of the server farm will affect the information technology application [68,69], and it is also believed that large-scale farms are more likely to adopt MIT. Consequently, the following research hypothesis was made:

**Hypothesis 3a.** *(Farm size): Large farm farmers are more likely to adopt the MIT.*

Internet-based regional obstacles are usually the outcome of the geographic environment of digital communication infrastructure and the absence of digital connection [26]. Regarding agricultural and IT adoption [60,63,67], evidence showed that the regional farm location is strongly related to higher IT adoption rates in the United States and Greece. About MIT, Matassa et al. [70] indicated that due to the decentralized digital infrastructure, the adoption of MIT could be influenced by the location of farm and agricultural activities. According to the information specified via the government of Pakistan, the MIT coverage in northwestern Pakistan seems to be relatively underdeveloped. Considering all those facts, the following hypothesis was assumed:

**Hypothesis 3b.** *(Region): The farm's location in the northwestern area is adversely associated with MIT adoption.*

The impact of farm diversification on agricultural technologies, including computers and the internet, has shown various interesting findings. For example, Alvarez et al. [71] and Roco [72] found no connection between farm diversification and equipment adoption, while Sekabira and Qaim [64] found a negative correlation between farm diversification and the adoption of IT. According to Roco [72], the diversification of farms is positively related to IT and MIT adoption. The owners of relatively diversified farms must gain more knowledge to make agricultural decisions, hence more likely to MIT adoption. According to the findings of Kaila and Tarp [73], diversified companies/farms have higher requirements for information technology. Although the outcomes obtained are different in the past study, we assumed the following hypothesis:

**Hypothesis 3c.** *(Farm diversification): Farms diversification is correlated to MIT adoption.*

## 4. Material and Methods

### 4.1. Model Specification

The dependent variables can be divided into two different sections. The first section is the MDs adoption (selection stage), which is the binary result of deciding whether the farmer adopts MDs that support the internet ($y_1 = 1$; otherwise, $y_1 = 0$); the second section (outcome stage) is the binary result, which determines whether the farmer adopt MIT ($y_2 = 1$; otherwise $y_2 = 0$). The probit method is a common econometric model which contains dependent variables and obtains binary results through maximum probability assessment [74]. By estimation, the two probit models for MDs adoption and MIT disregard the apparent association between the two. The bivariate probit method considers this association, which expands the probit technique [75]. However, the bivariate probit approach does not fully consider the predictable association between MDs adoption and MIT. More specifically, the MDs' adoption ultimately determines the probability of adoption of the MIT. Therefore, the selection stage result can only be observed if the farmers adopt the MDs. Hence, selection of sample bias may appear if the second result observation is not a random sample from the population [76].

The econometric model used to solve this issue is the bivariate probit method for sample selection. This model is based on the well-known Heckman selection model [77]. Alemi et al. [78,79] utilized the bivariate probit method with sample selection to examine smartphone use and mobile broadband adoption in Sweden. The following three types of outcomes were observed in this study [80]:

(i)     Wheat farmers do not adopt MDs ($y_1 = 0$);
(ii)    Wheat farmers adopts MDs but do not use MIT ($y_1 = 1$; $y_2 = 0$);
(iii)   Wheat farmers adopts MDs and use MIT ($y_1 = 1$; $y_2 = 1$).

Therefore, the following probabilities apply to all three categories of potential findings in the sample:

$$Y_1 = 0, \Pr(Y_1 = 0) = \Phi(-X_1\beta_1) \tag{1}$$

$$Y_1 = 1, Y_2 = 0, \Pr(y_1 = 1, y_2 = 0) = \Phi(x_1\beta_1) - \Phi_2(x_1\beta_1, x_2\beta_2, \rho) \tag{2}$$

$$Y_1 = 1, Y_2 = 1, \Pr(y_1 = 1, y_2 = 1) = \Phi_2(x_1\beta_1, x_2\beta_2, \rho) \tag{3}$$

The following log-likelihood function can be created by taking these probabilities into account:

$$\mathrm{InL} = \sum_{i=1}^{N} Y_{i1}Y_{i2}\mathrm{In}\Phi_2(X_1\beta_1, X_2\beta_2, \rho) + Y_{i1}(1 - Y_{i2})\,\mathrm{In}[\Phi(X_1\beta_1) - \Phi_2(X_1\beta_1, X_2\beta_2, \rho)] + (1 - Y_{i1})\,\mathrm{In}\Phi(-X_1\beta_1) \tag{4}$$

where Pr represents the probability that the farmer makes a binary decision, Y denotes the dependent variable of the selection and outcome equation, X denotes the independent variable vector for both farts, β is the estimated coefficient of each independent variable, and Φ is the unit normal distribution function. ρ represents the association between the two equation errors. Figure 2 shows the sample collection method of the proposed model.

According to Sartori [81], the instrumental variable is usually used to estimate the Heckman model in the selection phase, and this variable is not used in the outcome phase. No additional variables are required to determine the coefficients in the model. However, since the identification is based only on the parameter assumptions of bivariate normality and there are no missing variable deviations, Sartori [81] emphasizes that estimation techniques with similar descriptive variables are not recommended in the two equations.

Therefore, researchers have no other options other than to unblock an additional explanatory variable for the select equation or just classify it from the distribution hypothesis regarding the residuals [81]. In order to solve this problem, the outcomes of the proposed method are compared with the findings of the probit model, the bivariate probit model of the sample selection without instrumental variables, and the Sartori selection model [80,81].

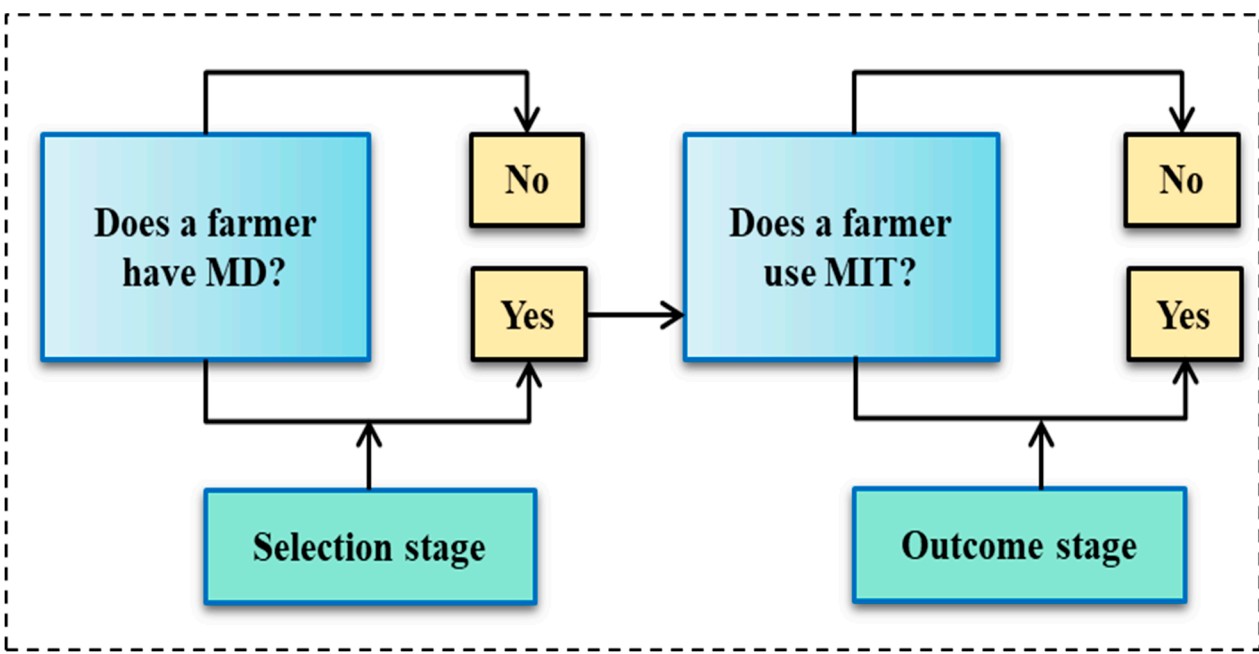

**Figure 2.** The bivariate probit model for sample selection for MIT adoption via wheat farmers. Note: MD = Mobile device; MIT = Mobile internet technology.

### 4.2. Study Area, Sampling Strategy, and Data Collection

#### 4.2.1. Wheat Production in the Study Area

Wheat is mainly produced in Punjab, Sindh, Balochistan, and the Khyber Pakhtunkhwa provinces of Pakistan. In this research, KPK province was selected as a study area. The study area has favorable climatic conditions for wheat and can produce high-quality wheat. However, wheat production in this region mainly relies on rainwater, and about 40% of farmers use irrigation. Small farmers, as cash crops, dominate wheat production. The sector needs modern technologies (e.g., MIT, MDs, and the Internet of Things) to support wheat production. However, in this study area, the mechanization rate is still relatively low, challenging wheat production. Wheat production is affected by pests and diseases globally, and Pakistan is no exception. Therefore, farmers usually replace wheat plants with new plants after 3–4 years of high yield to eliminate specific types of bugs that only attack wheat, better soil health and crop yield, etc. In 2020, wheat was planted at 772.3 thousand hectares of cropland in KPK and produced about 1400.5 thousand tons of wheat.

#### 4.2.2. Sampling Strategy and Data Collection

The study was conducted in KPK Province, Pakistan, from January to March 2021. In total, 650 questionnaires were distributed to the wheat farmers: a total of 628 questionnaires were considered fairly reliable to collect the data needed for this study, and 22 questionnaires were not considered in this study because they were not complete. This study used multistage random sampling techniques to collect the essential information from wheat farmers face to face (hard copy questionnaire distributed to the farmers in person). In order to understand the first stage of the MIT adoption by wheat farmers in KPK province, data were collected from four districts, namely, Dera Ismail Khan, Charsadda, Mansehra, and Swat, depending upon the share of agriculture production in these areas (Figures 3 and 4).

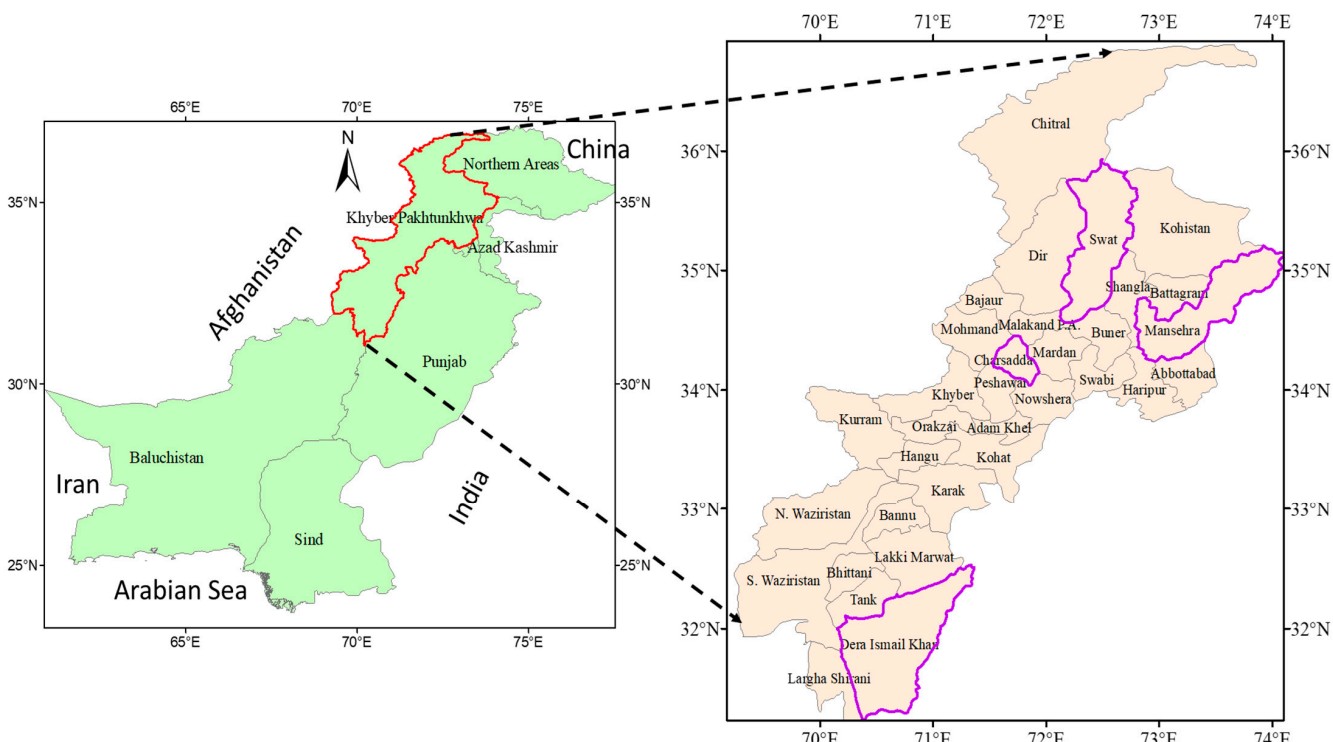

**Figure 3.** Map of the study area. The selected four district's boundaries are shown in pink color [37].

In the second stage, one tehsil was selected to fill out questionnaires, and in the third stage, one union council was targeted from each tehsil. In the fourth stage, four villages were focused randomly on each selected union council, and finally, the essential data were collected from wheat farmers from the studied villages. The questionnaires used in this study were divided into different parts. The first portion of the organized questionnaire contained the demographic and socio-economic characteristics of the respondents. The rest of the questionnaire aimed to obtain information about the MIT adoption by wheat farmers. The questionnaire was originally written in English and later translated into Urdu for the ease of the interviewees.

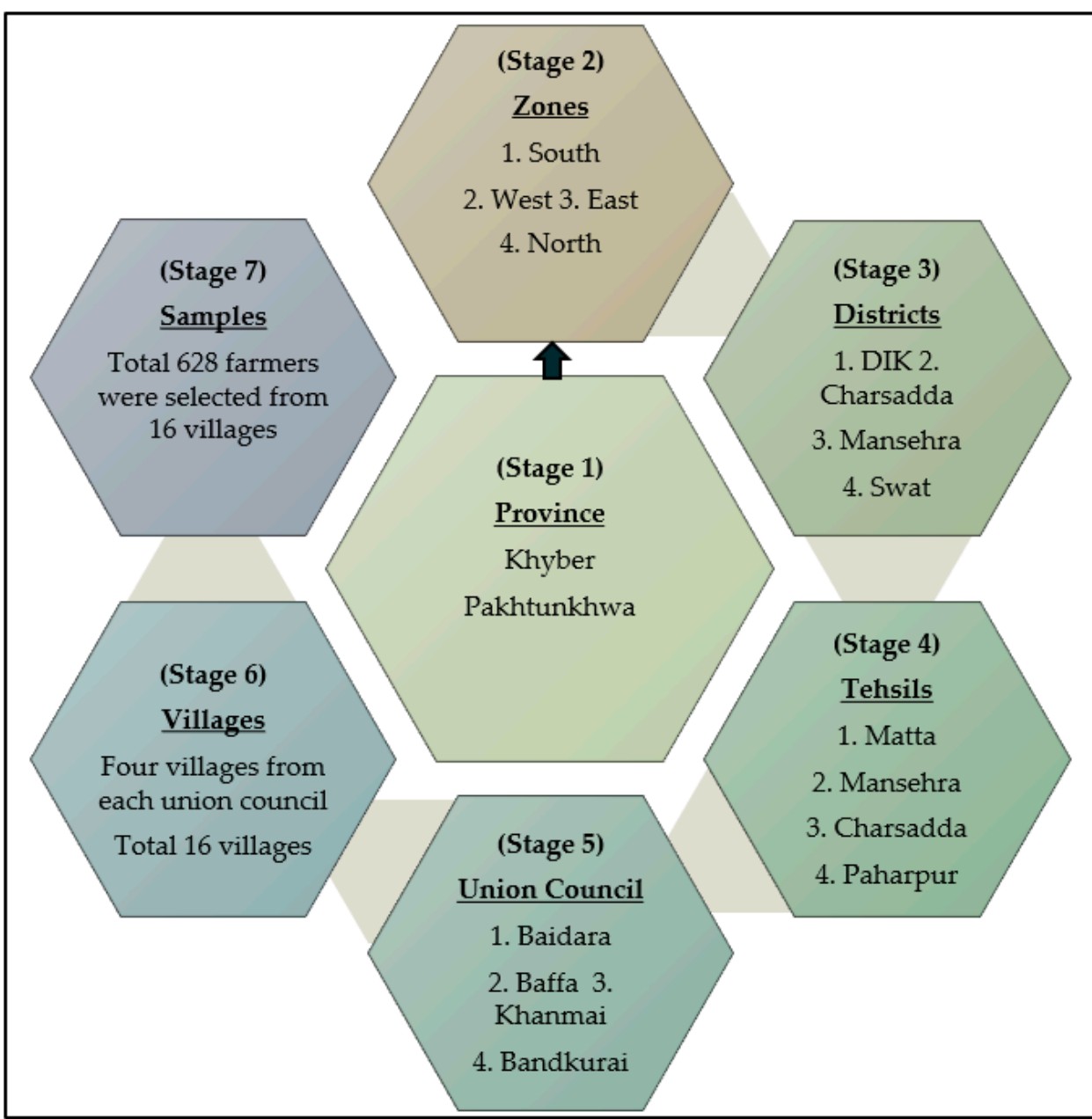

**Figure 4.** Sample distribution.

## 5. Results and Discussion

Table 1 indicates a statistics summary of the entire variables in the econometric assessment. On average, 65% of wheat farmers have MDs supporting these Internet technologies, and 55% use MIT in agricultural environments. In 2021, 61.64 million of the Pakistani population have accessed MIT via smartphones and/or tablets, which is less similar to the proportion of Pakistani farmers who use MIT through such MDs. Regarding sociodemographic variables, the average age of the respondents was 47.88 years old, 88% were males, and 17% of farmers had university degrees. The farm area varies from 5 to 20 ha, with an average of 7.17 ha. Table 1 also includes data regarding the area distribution of farms across Pakistan. For example, 24% of farms are located north of KPK. The farmers were also queried to utilize an isometric five-point Likert scale to express agreement or disagreement with two declarations. In order to measure innovation, farmers were questioned whether they were interested in using modern technologies or innovations, and on average, they slightly refused this (2.26%). Next, the interviewees were asked whether they had

enough knowledge to avoid harm to IT; on average, they also slightly rejected it (2.70%). In accordance with the framework of Adapa [82], the frequency of fixed IT usage in the MD adoption step (not in the result step) was calculated to illustrate the sample selection technique introduced in the prior portion. Moreover, 74% of the farmers surveyed use the fixed internet every day.

**Table 1.** Variables descriptions and descriptive statistics.

| Variables | Hypothesis | Explanation of Research Hypothesis | Mean (SD) |
|---|---|---|---|
| **Dependent variables** | | | |
| MIT | | 1 if farmers use MIT; 0 otherwise | 0.55 (0.04) |
| MD | | 1 if the farmer has Internet-enabled MD; 0 otherwise | 0.65 (0.02) |
| **Independent variables** | | | |
| Age | Hypothesis 1a | Age of the farmers (years) | 47.88 (11.77) |
| Gender | Hypothesis 1b | 1 if the farmer is male; 0 otherwise | 0.88 (0.03) |
| Education | Hypothesis 1c | 1 if the farmers hold a university degree; 0 otherwise | 0.17 (0.03) |
| Innovativeness | Hypothesis 1d | Once a new technological innovation arrives on the market, I will be interested in assessing it | 2.26 (1.07) |
| Awareness of IT risks | Hypothesis 2 | Aware to avoid the IT risks | 2.70 (1.12) |
| Farm size | Hypothesis 3a | Farm size (ha) | 7.17 (6.01) |
| Region | Hypothesis 3b | | |
| North | | The farmhouse is situated in the north of KPK | 0.24 |
| West | | The farmhouse is situated in the west of KPK | 0.26 |
| South | | The farmhouse is situated in the south of KPK | 0.42 |
| East | | The farmhouse is situated in the east of KPK | 0.07 |
| Farm diversification | Hypothesis 3c | Measurement of the farmhouse diversification | 0.25 (0.22) |
| **Control variable** | | | |
| IT usage (Regular) | | 1 if farmer use IT regularly; 0 otherwise | 0.74 |

**Note.** Standard errors (S.E.), Mean, and SD is shown as the ratio for MDs = 1.

Table 2 displays the evaluation results of the bivariate probit method for sample selection. Software version STATA 14 was used for analysis. The selected model Wald test was statistically significant ($p < 0.01$) and rejects the null hypothesis of simultaneous equality to zero of selected coefficients. The likelihood ratio test of $p = 0$ was rejected at the 1% significance level. The outcome shows that it was necessary to use the sample selection method.

The outcomes of the selection phase indicate that farmers who utilize the fixed IT daily are more likely to use MDs that support IT because the coefficient is statistically substantial and positive. This is consistent with the outcomes of Adapa [82]. In addition, young farmers with university degrees are more likely to use/adopt MDs. Additionally, similar outcomes were obtained by Hou et al. [34] for the adoption of tablets and smartphones. No association was found between the adoption rate of MDs and gender, farm size, and farm diversification.

**Table 2.** Outcomes of the bivariate probit model, including sample selection of MDs and MIT adoption for wheat farmers.

| Variables | Hypothesis | MD Adoption (SS) | MIT Adoption (OS) |
|---|---|---|---|
| | | Coefficient (S.E) | Coefficient (S.E) |
| IT use (Regular) | - | 0.7802 *** (0.0986) | - |
| Age | Hypothesis 1a | −0.0218 *** (0.0044) | −0.0201 *** (0.0054) |
| Gender | Hypothesis 1b | 0.0202 (0.1578) | 0.0617 (0.1677) |
| Education | Hypothesis 1c | 0.3177 ** (0.1335) | 0.0299 (0.1347) |
| Innovativeness | Hypothesis 1d | - | 0.192 8 *** (0.0485) |
| Awareness of IT risks | Hypothesis 2 | - | 0.0955 ** (0.0463) |
| Farm size | Hypothesis 3a | 0.033 (0.015) | 0.010 * (0.005) |
| Region | Hypothesis3b | | |
| North | | - | 0.3344 *** (0.1277) |
| West | | - | 0.4798 *** (0.1263) |
| South | | - | 0.3135 (0.1134) |
| East | | - | 0.0740 (0.2090) |
| Farm diversification | Hypothesis 3c | −0.1066 (0.2106) | −0.1102 (0.1347) |
| Constant | | 0.8708 *** (0.2890) | 0.4165 (0.3383) |
| Atanh ($p$) | | - | −1.3407 *** (0.2809) |
| $p$ | | - | −0.8718 (0.0674) |
| Likelihood ratio test for $p = 0$ | | 21.42 *** | - |
| Wald $x^2$ | | 42.55 *** | - |
| Log-likelihood | | −777.88 | - |

**Note:** SE, standard error, * $p < 0.10$, ** $p < 0.05$, and *** $p < 0.01$. (SS) denotes the selection stage, and (OS) outcome stage.

### 5.1. Study Hypothesis 1a–d

Figure 5 summarizes the hypothesis test outcomes. H1a indicates the age impact on the adoption of MIT, and the coefficient has a negative sign in statistics, indicating that, ceteris paribus, the older age is negatively correlated with the MIT adoption. Therefore, H1a could not be rejected. This outcome is consistent with the aforementioned research on MIT and the internet in sustainable agriculture development. Young farmers may be more interested in the use of innovative technologies, as [83,84] also pointed out. In addition, skills used in conjunction with information technology and MDs are generally well among young farmers [85]. It is consistent with the previous findings. For example, Woodburn et al. [86] found that older farmers have less computer and smartphones experience. However, young farmers have less agricultural experience [87]. Young farmers may use MIT as a source of additional knowledge for decision making. In short, young farmers are more likely to adopt MIT.

H1b assumes gender variances in the MIT adoption among wheat farmers. The measurement has anticipated indication but is not statistically positive at the 10% significance level. Therefore, the analysis cannot support H1b that gender is associated with MIT adoption. This is consistent with the regression outcomes of Yang et al. [88]. Although the past study shows that men are generally enthusiastic about modem technologies [53,89], especially mobile phones [90], their research shows that by comparing the statistics on the adoption of MIT over time, the gender difference is narrowing quickly [50]. By comparing the regression outcomes in 2017 and 2019, it was found that the statistical indication of gender is no longer significant for MIT adoption. This may describe the fact that there is also no association between the farmer's gender and the MIT adoption in the study. Therefore, men and women farmers have equal opportunities in the MIT adoption.

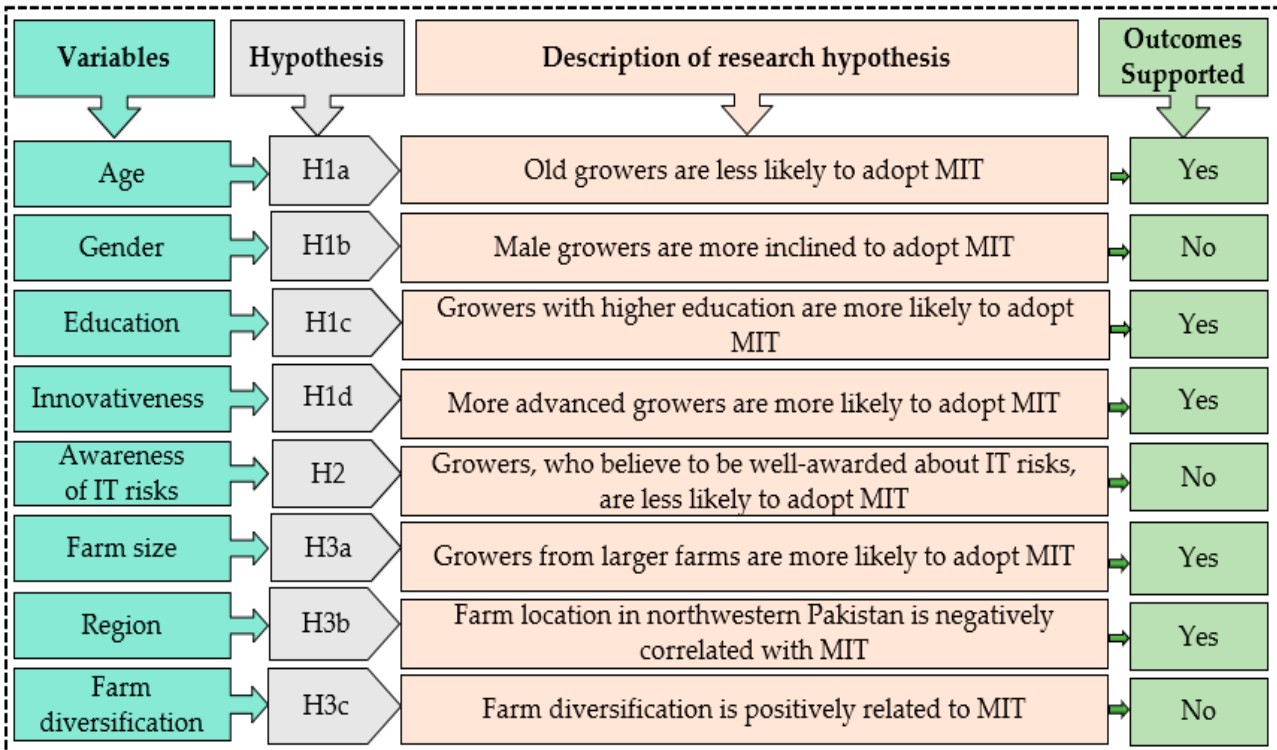

**Figure 5.** Explanation of the research hypothesis assessment outcomes.

According to H1c, the impact of farmer's education on MIT adoption was verified. The coefficient has an anticipated indication, but it is significant at the five percent level of significance. Therefore, the research cannot confirm H1c, which is that farmer's schooling is associated with the adoption of MIT (ceteris paribus). The outcomes are consistent with Islam et al. [91] and Khan et al. [92], but contrary to the previous research results, the latter shows an optimistic correlation between schooling and MIT adoption. However, as presented in the third column of Table 2, there is a significant correlation between education and MDs adoption. Therefore, if a farmer can generally utilize a smartphone or tablet due to his/her formal schooling, he/she already has the right to use the MIT. Furthermore, the educational impact on MIT adoption is possible because education makes it easier for farmers or anyone to process information [93]. In addition, well-educated farmers may also have more requirements for modern information [94]. Therefore, farmers with higher education can make more use of MIT to collect information. However, according to our results, education and MIT adoption are highly related.

H1d tested the impact of farmer's innovation on MIT adoption. Predictably, innovative farmers with ceteris paribus are more likely to adopt the MIT, because of the positive coefficient and has high statistical significance. Therefore, the analysis cannot reject H1d. These outcomes are consistent with the results of Hoang et al. [95]. Their outcomes indicate that the innovation of farmers' self-reports is significantly associated with their optimistic attitudes to mobile use [96]. This is also consistent with research on the use of technologies other than agriculture [97], because innovators adopt/use the latest/advanced skills and products faster than others. Therefore, innovative farmers adopt MIT faster than other non-innovative farmers.

### 5.2. Study Hypothesis 2

H2 shows the perception of farmers being informed regarding the dangers of IT. This coefficient has no estimated positive indicator and is statistically substantial. Therefore, the analysis could not deny H2 that farmer's awareness of IT danger (e.g., cyberbullying) is associated with MIT adoption (ceteris paribus). While it may not cause monetary loss,

infringement of user privacy is still the main concern of numerous internet users [98]. In addition, most farmers use the Internet and its applications for private purposes and primarily for commercial reasons [20,67]. Hence, they can process not only personal data but also commercially associated data that is considered very sensitive on IT. Intuitively, farmers can foresee the dangers of IT and therefore do not want to adopt/utilize MIT because they want to preserve their privacy and maintain the security of business-related information. The outcomes of our research can be elucidated as follows: The well-informed farmers may also understand how to develop suitable methods to safeguard their privacy when utilizing MIT and are thus more likely to adopt MIT.

For example, educated or informed farmers screen apps or websites before using the certificates. It has been indicated that certificates can support individuals, thereby raising the likelihood of online acquisitions [99]. Fecke et al. [100] also indicated that agribusiness utilizing e-commerce would consider ascertaining seals or certificates to upsurge trust. In addition, the outcomes also indicate that digital training courses for farmers would elucidate risks of internet use and how to develop suitable security instruments to promote farmers to use mobile information services.

### 5.3. Study Hypothesis 3a–c

H3a–c addresses the impact of farm characteristics on MIT adoption, and the farm size is significantly associated with the adoption of MIT. However, the coefficient has an optimistic indication and is statistically substantial at a significant level of 10%. Therefore, the research could not refuse H3a, which is that the size of the farm is related to MIT adoption. Large farms may face more multi-faceted decision making and higher complexity of the organization [101]. Thus, the MIT could be utilized to establish robust strategies for the effective business of the farm, such as banking and the acquisition of operating resources for sustainable agriculture development. In addition, it is possible to contact employees and consultants through an MIT-based messenger facility [67,102], which also supports the fact that farmers having huge farms may have a critical need for the latest technologies. Therefore, farmers on large farms may use MIT to gather data quickly. Specifically, the MIT allows farmers to obtain the price and weather information that is varied both in location and time. In summary, farmers who manage large farms have a higher chance of using the MIT.

It is estimated that the location of the farm is associated with MIT adoption verified with H3b. After estimation, the joint significance test shows that the coefficient of farm location is equivalent to zero. The assessment is statistically substantial ($x2$ (3) = 16.17, $p < 0.01$) and rejects the assumption that zero-coefficient is statistically insignificant. Therefore, the study could not refuse H3b. The northwestern region was set as a basic category in the initial econometric examination. The model expresses no statistically substantial variance between farmers whose farmhouses are situated in the eastern and northwestern regions. However, compared to the north and west fin, southern Pakistan is less likely to adopt MIT. Roco [72] proposed that the difference in digital infrastructure is the reason for the behavioral change in MIT adoption.

In addition, Bellon-Maurel et al. [103] indicated that the location could be understood as a proxy for internet access. Regarding the coverage of mobile broadband, the information provided by the current study shows that the coverage and long-term evolution of general mobile telecommunications services in the east of KPK are much smaller than in other regions, which may explain the results. In addition, cultural variances (for example, most of the farmers in Pakistan are more conservative) may become an obstacle to adopting innovations such as the MIT. Though this study did not explicitly consider this dimension, it can be inferred that the farm location will affect the adoption of MIT.

Finally, H3c addresses the farm diversification influence on MIT adoption, and the selected model shows that farm diversification is not correlated to MIT adoption. Therefore, this research could not facilitate H3c that farm diversification is significantly related to the adoption of MIT. This outcome is consistent with Roco [72] view on the adoption of

computers. However, diversified farms may have a higher need for skills and MIT usage for several production functions and information collection. It is conceivable that there is a positive correlation between farm diversification and the use of MIT. However, the outcomes show that MIT adoption has nothing to do with the diversification of farms.

## 6. Conclusions

The study analyzed a representative data set of 628 wheat farmers to understand the MIT adoption and use in agriculture development. This study used the bivariate probit method for sample selection to examine the crucial factors influencing MIT adoption. The outcomes indicate that farmer's age (H1d) and farm's size (H3a) correlate with MIT adoption. In addition, the study findings suggest that farm location (H3b) is related to farmer's MIT adoption rate. Additionally, educated (H1c) and innovative (H1d) farmers are most likely to adopt MIT. However, the outcomes showed gender (H1b), awareness of IT risks (H2), and farm diversification (H3c) are not correlated with the MIT adoption.

Farmers who fully understand the IT danger are more inclined to adopt MIT, which is counterintuitive. The fact that well-informed farmers may have put safety measures to deal with possible risks may explain this result. The implication of this result has two aspects: the apprenticeship should include the digital aspect so that farmers are aware of the potential risks of IT use. Correspondingly, information on measures to ensure online security will also be provided. In addition, agricultural equipment providers that depend on MIT (such as smart agricultural technology) should know farmer's safety issues and strive to clarify the dangers related to MIT to decrease unwillingness in the adoption process. This may also be achieved by providing a certificate or stamping.

The innovations of large farms and young farmers are the focus groups of marketing pursuits because they are expected to become MIT adopters. For example, suppliers and providers of cutting-edge and modern agricultural technology can focus on marketing mobile phones to achieve their focus group. These findings indicate that the young farmers of large farms are expected to utilize MIT. Therefore, providers can emphasize the possibility of integrating MDs and IT with these tools for this focus group. Moreover, this research benefits the farmers to increase their agricultural production by adopting MIT, which ultimately enhances their skills for modern agriculture. On the other hand, this study also benefits policymakers to understand the advantage of MIT for modern agriculture and their role in supporting farmers through IT infrastructure development. However, policymakers of this region/country should consider expanding the coverage of MIT in remote areas. The results also highlight the need for wheat farmers to have MIT services in the study area. To gain more insights, consider the farmer's location and satisfaction with mobile broadband coverage, not just the farm's location.

This study has some limitations. Firstly, the current research was conducted during the COVID-19 pandemic issues. Secondly, due to financial concerns, this study only focused on four districts in the KPK province of Pakistan. Hence, an inclusive study is needed in future research. The data essentially cover one province in Pakistan, so it is difficult to generalize the conclusions at the national level. Therefore, future studies should use more representative samples, which may have wider implications for rural Pakistan. Possible research must examine how and to what extent MIT use promotes farmer's income and market participation to facilitate a supply chain asset and financial efficiency perspective. However, this study provides different starting points for other research projects. For instance, this research can be applied to other developed and developing countries. An in-depth analysis of farmer's awareness and familiarity with specific MIT risks, such as phishing, should also be carried out. It is also worth examining how farmers specifically integrate MIT and related internet content or associated applications into their farm business responsibilities. Additionally, it may be interesting to see how mobile and fixed internet technologies differ in retrieving farm business-related information.

**Author Contributions:** N.K., R.L.R., H.S.K. and S.Z. developed and outlined this concept, including method and approach to be used; N.K. and R.L.R., developed and outlined the manuscript; N.K. and S.Z. contributed to the methodology and revision of this manuscript; N.K. and R.L.R. wrote the article. All authors have read and agreed to the published version of the manuscript.

**Funding:** The authors extend their appreciation to the researchers supporting the project of (No: RSP-2021/403) King Saud University, Riyadh, Saudi Arabia.

**Institutional Review Board Statement:** Not applicable.

**Informed Consent Statement:** Not applicable.

**Data Availability Statement:** The data that support our research findings are available from the corresponding author on request.

**Conflicts of Interest:** The authors declare no conflict of interest.

## Abbreviations

| | |
|---|---|
| MIT | Mobile Internet Technology |
| MPU | Mobile Phone Usage |
| BPM | Bivariate Probit Method |
| ICTs | Information Communication Technologies |
| IT | Internet Technology |
| AM | Agricultural Modernization |
| H | Hypothesis |
| PAK | Pakistan |
| RH | Research Hypothesis |
| MDs | Mobile Devices |
| SS | Selection Stage |
| OS | Outcome Stage |
| KPK | Khyber Pakhtunkhwa |
| MITA | Mobile Internet Technology Adoption |
| DIK | Dera Ismail Khan |
| UC | Union Council |

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
