# Peer review of "Mobile Internet Technology Adoption for Sustainable Agriculture: Evidence from Wheat Farmers"

_applsci, doi:10.3390/app12104902_

Round 1

Reviewer 1 Report

The author has presented an overview of the adoption of mobile internet technology among farmers in Pakistan and presented a statistical analysis.

  • The most important thing Pakistan is a developing country where farmers have highly poor knowledge regarding technology. So, the adoption of MIT in farming is a baseless review, due to the poor literacy rate and lack of trust as well as high-cost infrastructure farmers don’t prefer to move from traditional farming methods towards innovative solutions.
  • Which questionnaire scale has been followed by the authors for their analysis?
  • What is the main motivation behind this manuscript because there is a large number of smart farming technologies available which are revolutionizing the agriculture industry in more innovative ways such as robotics, IoT, blockchain, etc.?
  • There is a large number of grammatical errors and typo mistakes.
  • What are the challenges and gaps with the adoption of MIT in agriculture?
  • What are the future suggestion which author wants to give to technologists, researchers, agriculturists, and farmers?
  • The collected data for analysis is not enough and sufficient to conclude effective and fruitful results. The author should consider more areas such as Punjab is the largest province of Pakistan as well as the top agriculture perspective.
  • Moreover, the reviewer is not able to understand the novelty and contribution of the work. This contribution is not enough to consider for publication. Because selected areas and collected data is very limited.
  • Resubmission is allowed after making suggested changes. In order to submit author should consider more data and areas for analysis and add challenges as well as future directions.

Author Response

Dear Reviewer,

We are very grateful for your valuable time and your constructive comments and suggestions to help improve the quality of the manuscript. We have improved the manuscript according to your suggestions.

Reviewer 2 Report

I propose to update, if possible, references older than 10 years. There are 18 such items.

I propose to omit 8 items from the literature that are self-citations.

Author Response

Dear Reviewer,

We are very grateful for your valuable time and your constructive comments and suggestions to help improve the quality of the manuscript. We have improved the manuscript according to your suggestions.

Thank you
